# INDIVIDUAL FAIRNESS OF DATA PROVIDER REGARDING PRIVACY RISK AND GAIN

## ABSTRACT

Fairness and privacy risks are important concerns of machine learning (ML) when deploying ML to the real world. Recent studies have focused on group fairness and privacy protection, but no study focuses on individual fairness (IF) and privacy protection. In this paper, we propose a new definition of IF from the perspective of privacy protection and experimentally evaluate privacy-preserving ML based on the proposed IF. For the proposed definition, we assume that users provide their data to an ML service and consider the principle that all users should obtain gains corresponding to their privacy risks. As a user's gain, we calculate the accuracy improvement on the user's data when providing the data to the ML service. We conducted experiments on the image and tabular datasets using three neural networks (NNs) and two tree-based algorithms with differential privacy guarantee. The experimental results of NNs show that we cannot stably improve the proposed IF by changing the strength of privacy protection and applying defenses against membership inference attacks. The results of tree-based algorithms show that privacy risks were extremely small without depending on the strength of privacy protection but raise a new question about the motivation of users for providing their data.

## 1 INTRODUCTION

As machine learning (ML) services trained with users' data become increasingly popular, privacy risks of memorizing training data have been gaining attention (Shokri et al., 2017; Jagielski et al., 2020; Nasr et al., 2021; Malek Esmaeili et al., 2021). To prevent privacy leakage through trained models, privacy-preserving ML based on differential privacy (DP) (Dwork et al., 2006) is a de facto standard. For example, DP-SGD (Song et al., 2013; Abadi et al., 2016) is used for training neural networks (NNs) based on stochastic gradient descend (SGD) with DP guarantee, and DPBoost (Li et al., 2020) and DPXGBoost (Grislain & Gonzalvez, 2021) are used for training tree-based models with DP guarantee.

When applying ML to the real world, fairness is another important concern about ML. Recent studies have begun to focus on both privacy protection and fairness: the difference in the effect of DP on majority and minority groups (Bagdasaryan et al., 2019; Pujol et al., 2020; Farrand et al., 2020; Tran et al., 2021), the difference in vulnerabilities against membership inference attacks (MIAs) between majority and minority groups (Zhang et al., 2020; Zhong et al., 2022), and methods for guaranteeing both group fairness and DP (Xu et al., 2019; 2020). All of these studies have focused on group fairness, i.e., fairness between majority and minority groups. Assuming situations where users decide whether to provide their data to ML services, individual fairness (IF), i.e., fairness between individual users, is also important for the decision. However, no study has focused on IF and privacy protection.

In this paper, we investigate privacy-preserving ML from the perspective of both IF and privacy protection. To this end, we propose a new definition of IF from the perspective of privacy protection and experimentally evaluate privacy-preserving ML based on the proposed IF. Assuming that users provide their data to an ML service, we define the proposed IF based on the principle that all users should obtain gains corresponding to their privacy risks. Furthermore, we discuss the relationship between the proposed IF and prior IF for classification and validate the proposed IF using synthetic data.

We extensively evaluate privacy-preserving ML in terms of the proposed IF. Using two image datasets, we evaluate a six-layer convolutional NN (CNN) and ResNet18 (He et al., 2016) trained with DP-SGD (Song et al., 2013; Abadi et al., 2016). Using two tabular datasets, we evaluate a five-layer fully connected NN trained with DP-SGD, DPBoost (Li et al., 2020), and DPXGBoost (Grislain & Gonzalvez, 2021). In the evaluation, as a user's privacy risk, we calculate a lower bound of a DP parameter $\epsilon$ (Jagielski et al., 2020; Malek Esmaeili et al., 2021). As a user's gain, we calculate the accuracy improvement on the user's data when providing the data to the ML service. Since the accuracy improvement means that the utility of the ML service increases for the user, we can regard the accuracy improvement, i.e., the utility increase, as the user's gain.

The results were different for NNs and tree-based algorithms. The main findings are as follows.

- The results of NNs show that unfairness in terms of the proposed IF was large without depending on the strength of privacy protection because some users' gains were small compared with their privacy risks. These results show that we cannot improve the proposed IF by adjusting the strength of privacy protection.

- We further evaluated the proposed IF when applying defenses against MIA to NNs. No defense improved fairness without depending on the settings (i.e., datasets, NNs, and strength of privacy protection), and fairness was degraded by the defenses in some settings. These results show the need for a method that stably improves the proposed IF of NNs without depending on the settings.

- The results of tree-based algorithms show that privacy risks and gains were extremely small without depending on the strength of privacy protection. For tree-based algorithms, the proposed IF does not seem to be important, but these results raise a new question about the motivation of users for providing their data. For example, some users are unwilling to provide their data if their data do not improve the ML service.

## 2 PRELIMINARIES

**Individual fairness.** IF is a main concept of algorithmic fairness along with group fairness. IF is proposed for the classification task based on the principle "similar data should be classified similarly" (Dwork et al., 2012). Let input space be $V$, a set of output classes be $A$, probability distributions over output classes be $\Delta(A)$, a mapping from an input to an output, i.e., an ML model, be $M : V \to \Delta(A)$, and distance in input and output space be $d : V \times V \to \mathbb{R}$ and $D : \Delta(A) \times \Delta(A) \to \mathbb{R}$. If an model is a Lipschitz mapping, the model satisfies the principle of IF.

**Definition 1** (Lipschitz mapping). A mapping $M : V \to \Delta(A)$ satisfies the $(D, d)$-Lipschitz property if for any $x, y \in V$, the following holds:

$$D(M(x), M(y)) \le d(x, y).$$

$d$ and $D$ need to be designed for each task. An example of $d$ is a Mahalanobis distance without using features correlated with sensitive attributes such as races and genders.

Another definition based on the same principle is proposed by relaxing the Lipschitz property.

**Definition 2** ($\epsilon$-$\delta$-IF (John et al., 2020)). A mapping $M$ is $\epsilon$-$\delta$-individually fair if for all $x, y$ such that $d(x, y) \le \epsilon$, the following holds:

$$|M(x) - M(y)| \le \delta.$$

Note that in practice, a task-specific loss needs to be considered in addition to these definitions of IF for building a fair and accurate model.

**Differential privacy.** DP (Dwork et al., 2006) is a standard definition of privacy protection for statistical data analysis. In DP, we consider neighboring datasets $D_0$ and $D_1$ differing by only one sample. An example is adding one sample $(\boldsymbol{x}', y')$ to $D_0$ to make $D_1$, i.e., $D_1 = D_0 \cup \{(\boldsymbol{x}', y')\}$.

**Definition 3** (Differential Privacy). A randomized mechanism $\mathcal{M} : \mathcal{D} \to \mathbb{R}$ is $(\epsilon, \delta)$-differentially private if for any neighboring datasets $D_0, D_1$ and for any output range $S \subset \mathbb{R}$, the following holds:

$$Pr[\mathcal{M}(D_0) \in S] \le e^\epsilon Pr[\mathcal{M}(D_1) \in S] + \delta.$$

A small constant, e.g., $10^{-5}$, is typically used for $\delta$, and $\epsilon$ represents the privacy risk of the mechanism. Privacy-preserving ML guarantees an upper bound of $\epsilon$ with theoretical analysis.

**Lower bound of DP parameter $\epsilon$.** From the practical perspective, the lower bound of the DP parameter $\epsilon$ is empirically studied by instantiating attackers against privacy-preserving ML (Jagielski et al., 2020; Nasr et al., 2021; Malek Esmaeili et al., 2021). We explain the basic idea of these methods following the prior work (Jagielski et al., 2020). This method is based on a game of an attacker and trainer shown in Fig. 1. First, the attacker prepares two neighboring datasets $D_0$ and $D_1$. Here we consider $D_1$ is prepared by adding a target sample $(\boldsymbol{x}', y')$ to $D_0$. Second, the trainer randomly selects one of the neighboring datasets $D_b$, where

Figure 1: Game for calculating a lower bound of $\epsilon$

$b$ is the index of the selected dataset. The trainer builds the model $f$ using $D_b$ with a privacy-preserving ML algorithm $\mathcal{M}$ and returns the loss $l = \ell(f(\boldsymbol{x}'), y')$ to the attacker. Third, the attacker predicts which dataset is used in training with an algorithm $A(D_0, D_1, l)$ and sends the prediction $b'$ to the trainer. Since a small loss indicates that the sample $(\boldsymbol{x}', y')$ is used in training, a commonly used algorithm predicts that the dataset is $D_1$ if the loss is less than the threshold. Finally, the trainer checks if the prediction is correct.

To calculate the lower bound, this game is repeated many times, e.g., 1,000 times. If $\mathcal{M}$ is $(\epsilon, \delta)$-differentially private, the false positive rate (FPR) and false negative rate (FNR) of the games are bounded (Kairouz et al., 2015) by $FPR + e^\epsilon FNR \leq 1 - \delta$ and $FNR + e^\epsilon FPR \leq 1 - \delta$.

Given $\delta$, the maximum $\epsilon$ satisfying the above inequalities is the empirical $\epsilon$. Its lower bound $\epsilon_{LB}$ is defined using the upper bounds of FPR and FNR calculated with the Clopper-Pearson method (Clopper & Pearson, 1934):

$$\epsilon_{LB} = \max\left(\log\frac{1 - \delta - FPR_{UB}}{FNR_{UB}}, \log\frac{1 - \delta - FNR_{UB}}{FPR_{UB}}\right). \tag{1}$$

Assuming that the target sample $(\boldsymbol{x}', y')$ is data provided by a user, we can estimate the user's privacy risk by $\epsilon_{LB}$.

## 3 INDIVIDUAL FAIRNESS FOR PRIVACY PROTECTION

We propose a new definition of IF from the perspective of privacy protection assuming that users provide their data to an ML service. Additionally, we discuss the relation between the proposed IF and the prior IF for the classification task and validate the proposed definition using synthetic data.

### 3.1 PROBLEM SETTING

Before describing the proposed IF, we explain the ML service and user that we assume in this paper.

**Machine learning service.** We assume a service using an ML model trained with users' data. The service accepts inputs from users and returns the predictions made by the model. In parallel with running the service, the service provider continues to collect users' data and add them to the training dataset. If different users have different traits of data, data needs to be collected for accurately predicting new users' data. Regarding data collection, we assume that users can select whether to provide their data to the service or not. For example, when they start to use the service, they are asked whether they consent to share their data. If they consent, their data will be shared with the service and added to the training dataset. Examples of the services are facial expression recognition, handwritten text recognition, product recommendation, and medical diagnosis.

**User.** We assume that users of the above ML service expect that their data are accurately classified by the service. The users decide whether to provide their data depending on the expected privacy

risks. If the users consider that the risks are high, they decide not to provide their data. If the users consider that the risks are low, they decide to provide their data.

The typical privacy risk is caused by the memorization of the ML model. The memorization leaks information on the training dataset from the outputs of the model. Such privacy risk is empirically measurable by calculating the lower bound of DP parameter $\epsilon$ as described in Section 2. In this paper, we define IF regarding privacy protection when users provide their data.

### 3.2 Proposed Definition of Individual Fairness

We define IF between users who provide their data to the ML service from the perspective of privacy protection. Since a privacy risk, i.e., the lower bound of $\epsilon$, can be calculated with the method described in Section 2, IF regarding the privacy risks could be defined by referring to IF for the classification task. Two naive definitions are as follows.

- A training algorithm is individually fair if users having similar data face similar privacy risks. This definition is inadequate because a difference in privacy risks can be large if users' data are dissimilar. For example, users having outlier data may have to tolerate high risks despite the fact that users having ordinary data face low risks.

- A training algorithm is individually fair if the difference in privacy risks between any pair of users is small. This definition can be satisfied by reducing privacy risks with privacy-preserving ML because the differences are small if all users' privacy risks are small. However, strong privacy protection with DP is known to degrade classification performance (Abadi et al., 2016). For this reason, this definition seems to be impractical.

As described above, IF for privacy protection cannot be adequately defined in naive ways. To solve this problem, we admit that there are differences in privacy risks between users and define IF by additionally considering gains that users obtain in response to providing their data. We consider that a training algorithm is individually fair if all users obtain gains corresponding to their privacy risks. In other words, an algorithm is fair if users facing high privacy risks obtain large gains and if users facing low privacy risks obtain small gains. In contrast, an algorithm is unfair if users obtain much larger or smaller gains than expected from their privacy risks.

In this paper, we calculate a user's gain by the accuracy improvement on the user's data. Since the user expects that their data is accurately classified by the ML service, the accuracy on the user's data corresponds to the utility of the ML service. If the accuracy is improved by providing the data, we can regard the accuracy improvement, i.e., the utility increase, as the user's gain. Note that we discuss other types of gains in Section 5. We estimate the accuracy improvement using shadow models. We use a dataset $D_0$ already collected by the service and a user's sample $(\boldsymbol{x}_i, y_i)$. We train $n_s$ shadow models $\{f_{out}\}_{i=1}^{n_s}$ using $D_0$, i.e., without the user's sample, and $n$ shadow models $\{f_{in}\}_{i=1}^{n_s}$ using $D_0 \cup \{(\boldsymbol{x}_i, y_i)\}$, i.e., with the user's sample. Then we count $c_{out}$ and $c_{in}$: the number of shadow models accurately predicting the user's sample $(\boldsymbol{x}_i, y_i)$ among $\{f_{out}\}_{i=1}^{n_s}$ and $\{f_{in}\}_{i=1}^{n_s}$, respectively. The accuracies on the sample $(\boldsymbol{x}_i, y_i)$ when models are trained without and with $(\boldsymbol{x}_i, y_i)$ are estimated by $\frac{c_{out}}{n_s}$ and $\frac{c_{in}}{n_s}$. The accuracy improvement, i.e., the gain, is calculated by $g_i = \frac{c_{in}}{n_s} - \frac{c_{out}}{n_s}$. Note that we use the accuracy improvement instead of the loss decrease because the utility of the ML service does not increase even if the loss decreases unless the falsely predicted user's sample becomes accurately predicted. As the user's privacy risk $r_i$, we calculate the lower bound of $\epsilon$ using $(\boldsymbol{x}_i, y_i)$ as the target sample.

Using the above users' privacy risks and gains, we define IF for privacy protection. Since we admit the difference in privacy risks between users, we focus on the difference in the tradeoff of privacy risks and gains. Specifically, we consider that a training algorithm is individually fair if all users' tradeoffs are similar. We denote privacy risks and gains of users $U = \{u_i\}_{i=1}^{n}$ as $R = \{r_i\}_{i=1}^{n}$ and $G = \{g_i\}_{i=1}^{n}$, respectively. To evaluate the tradeoff, we normalize $R$ and $G$ so that their means and variances are 0 and 1, respectively. The normalized risk and gain of the user $u_i$ are denoted as $r_i' = \frac{r_i - \mu_r}{\sigma_r}$ and $g_i' = \frac{g_i - \mu_g}{\sigma_g}$, where $\mu_r$ and $\mu_g$ are means of $R$ and $G$, and $\sigma_r^2$ and $\sigma_g^2$ are variances of $R$ and $G$. In an ideally fair situation, gains are completely correlated with risks and all users' tradeoffs are the same. In this case, we have $g_i' = r_i'$ for all users. Based on this insight, we define IF by focusing on the difference between an ideal gain (i.e., $r_i'$) and actual gain (i.e., $g_i'$).

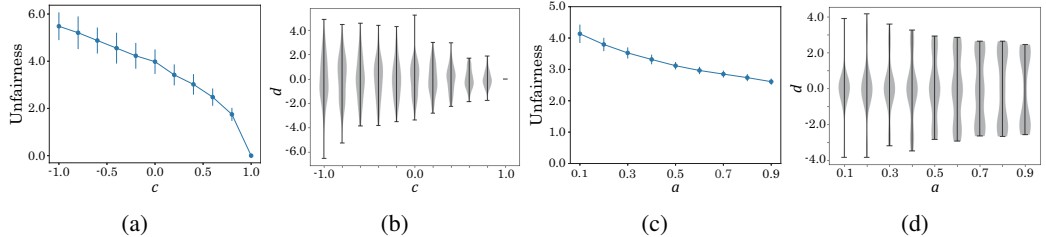

Figure 2: Unfairness and distribution of $d_i$. (a) and (c) show the mean and standard deviation of unfairness calculated 100 times. (b) and (d) show the distribution of $d_i$ calculated using 100 generated samples.

**Definition 4** ($\delta$-IF regarding privacy risks and gains). Let $R' = \{r'_i\}_{i=0}^n$ and $G' = \{g'_i\}_{i=0}^n$ be the normalized privacy risks and gains so that their means are 0 and variances are 1. A training algorithm is $\delta$-individually fair if for all users $u_i \in U$, the following holds:

$$|g'_i - r'_i| \leq \delta.$$

When $G'$ is completely correlated with $R'$, we have $\delta = 0$. $\delta$ is expected to increase as the correlation becomes weaker. When $G'$ is inversely correlated with $R'$, we have a large $\delta$. Note that even if most users have gains corresponding to their risks, if only one user has a small or large gain compared with the risk, we have a large $\delta$. This is because our definition is based on the worst-case user in the same way as Definition 1 and Definition 2 of IF for the classification task.

Based on this definition, we can evaluate the unfairness $\hat{\delta}$ of a training algorithm by using empirically calculated $R'$ and $G'$:

$$\hat{\delta} = \max_{u_i \in U} |g'_i - r'_i|. \tag{2}$$

Additionally, we can investigate a training algorithm in detail in terms of the proposed IF using each user's difference between the gain and risk: $d_i = g'_i - r'_i$.

### 3.3 RELATION TO $\epsilon$-$\delta$-IF

Here, we discuss the relation between the proposed $\delta$-IF and $\epsilon$-$\delta$-IF in Definition 2. $\epsilon$-$\delta$-IF guarantees that outputs of an ML model are similar if input data are similar. In the setting of this paper, we can consider $\epsilon$-$\delta$-IF guaranteeing that users' gains are similar if their privacy risks are similar. Specifically, we modify $\epsilon$-$\delta$-IF by replacing $x, y$ with $r_i, r_j$ and $M(x), M(y)$ with $g_i, g_j$. For the distance of privacy risks, we simply calculate the absolute difference of privacy risks: $d(r_i, r_j) = |r_i - r_j|$. If the modified $\epsilon$-$\delta$-IF is satisfied, for all users such that $|r_i - r_j| \leq \epsilon$, we have $|g_i - g_j| \leq \delta$. The proposed $\delta$-IF has a relation to the modified $\epsilon$-$\delta$-IF.

**Proposition 1.** If a training algorithm satisfies $\delta$-IF regarding privacy risks and gains, the algorithm also satisfies $\epsilon$-$\delta'$-IF in the setting where inputs are privacy risks and outputs are gains. Here, $\delta' = \frac{\sigma_g}{\sigma_r}\epsilon + 2\sigma_g\delta$.

For the proof, please refer to Appendix A. Note that the converse is not true because $\epsilon$-$\delta$-IF does not hypothesize the correlation between privacy risks and gains.

### 3.4 VALIDATION USING SYNTHETIC DATA

We validate the proposed IF using two synthetic data. As described above, unfairness $\hat{\delta}$ is expected to be small if $g'$ is correlated with $r'$, and $\hat{\delta}$ is expected to be large if $g'$ is inversely correlated with $r'$. We verify this expectation using the first synthetic data. Specifically, we generate $\begin{bmatrix} r_i \\ g_i \end{bmatrix}$ using a two-dimensional Gaussian distribution with a mean $\mu = \begin{bmatrix} 0 \\ 0 \end{bmatrix}$ and covariate matrix $\Sigma = \begin{bmatrix} 1 & c \\ c & 1 \end{bmatrix}$, changing $c$ from -1.0 to 1.0 by 0.2. For each $c$, we calculate $\hat{\delta}$ 100 times using 100 generated samples. We show examples of synthetic data in Fig. 5 in Appendix and unfairness $\hat{\delta}$ in Fig. 2(a). As expected, $\hat{\delta}$ was 0 when $g'$ was completely correlated with $r'$, and $\hat{\delta}$ was the largest when $g'$ was completely

inversely correlated with $r'$. When $g'$ did not correlate with $r'$, i.e., $c = 0.0$, $\hat{\delta}$ was close to 4.0. This result shows that a training algorithm is unfair if $\hat{\delta}$ is close to 4.0 or larger. Figure 2(b) shows the distributions of $d_i = g_i' - r_i'$. Since the Gaussian distribution generates many data close to the mean, the number of users having $d_i$ close to 0 was large. The maximum and minimum values of $d_i$ becomes large and small as $c$ decreases. Note that based on the modified $\epsilon$-$\delta$-IF, unfairness is small when $c = -1.0$ because users having similar $r'$ obtain similar $g'$. This is because the correlation between $r'$ and $g'$ is not assumed in the modified $\epsilon$-$\delta$-IF.

We generate the second synthetic data to confirm that the distribution of $d_i$ is useful. The second synthetic data consists of three types of users. The first type of users obtain $g'$ corresponding to $r'$, and we generate $\left[\begin{smallmatrix} r_i \\ g_i \end{smallmatrix}\right]$ using a two-dimensional Gaussian distribution with $\mu = \left[\begin{smallmatrix} 0 \\ 0 \end{smallmatrix}\right]$ and $\Sigma = \left[\begin{smallmatrix} 1 & 0.8 \\ 0.8 & 1 \end{smallmatrix}\right]$. The second type of users obtain small $g'$ compared with $r'$, and we generate $\left[\begin{smallmatrix} r_i \\ g_i \end{smallmatrix}\right]$ using a distribution with $\mu = \left[\begin{smallmatrix} 2 \\ -2 \end{smallmatrix}\right]$ and $\Sigma = \left[\begin{smallmatrix} 0.1 & 0 \\ 0 & 0.1 \end{smallmatrix}\right]$. The third type of users obtain large $g'$ compared with $r'$, and we generate $\left[\begin{smallmatrix} r_i \\ g_i \end{smallmatrix}\right]$ using a distribution with $\mu = \left[\begin{smallmatrix} -2 \\ 2 \end{smallmatrix}\right]$ and $\Sigma = \left[\begin{smallmatrix} 0.1 & 0 \\ 0 & 0.1 \end{smallmatrix}\right]$. We generate 100 samples by changing the ratio of unfair users from 0.1 to 0.9 by 0.1. When the ratio is $a$, we generate $(1 - a) \times 100$ samples using the first distribution, $a \times 50$ samples using the second distribution, and $a \times 50$ samples using the third distribution. Examples of the synthetic data are shown in Fig 6 in Appendix. For each ratio $a$, we calculate unfairness $\hat{\delta}$ 100 times as shown in Fig. 2(c). Since the generated data always contained unfair users, unfairness was large without depending on $a$. Even though unfairness is similar, Fig. 2(d) shows the difference in the distributions of $d_i$. When $a$ was small, the number of unfair users was small, and the majority of users had $d_i$ close to 0. When $a$ was close to 0.5, there were various types of users; $d_i$ was close to 0, large, and small. When $a$ was large, the number of unfair users was large, and the majority of users had large or small $d_i$.

# 4 EXPERIMENT

## 4.1 EXPERIMENTAL SETUP

We describe important points of the setup. For more details, please refer to Appendix B.

**Dataset.** We used two image datasets and two tabular datasets containing user information. The image datasets are FEMNIST and Celeba, both made by Caldas et al. (2018). FEMNIST contains 28×28 gray scale handwritten digits for 10 class-classification. Celeba contains 64×64 facial images for classifying "smile" and "not smile". The tabular datasets are Adult (Kohavi & Becker, 1996) and Texas (Texas Department of State Health Services, 2013). Adult contains users' attribute vectors of size 108 for classifying whether their income is larger than 50k. Texas contains patients' attribute vectors of size 72 for classifying whether their total charge is larger than 50k. For each dataset, we selected 100 users from the test datasets to evaluate fairness. We selected a variety of users from ordinary to outlier users to estimate the overall trend of the proposed IF. For the detailed procedure, please refer to Appendix B.

**Machine learning algorithm and hyperparameters.** We used two NNs for the image datasets (ConvNet and ResNet18) and one NN (FC) and two tree-based algorithms (DPBoost and DPXG-Boost) for the tabular datasets.

- *ConvNet*: This is a six-layer CNN designed by referring to the prior work (Nasr et al., 2021). ConvNet consists of two convolutional layers, two max-pooling layers, and two fully connected layers. We trained ConvNet with DP-SGD and changed the strength of privacy protection using different variances of the noises: $\sigma = 0.1, 0.3, 0.5$, and $0.7$.

- *ResNet18*: This is an 18-layer CNN with the shortcut connections (He et al., 2016). We trained ResNet18 with DP-SGD using the same hyperparameters as ConvNet.

- *FC*: This is a five-layer fully connected NN. The number of units in the intermediate layers is 500, and their activation functions are ReLU. We trained FC with DP-SGD using the same hyperparameters as ConvNet.

- *DPBoost*: This is a differentially private Gradient Boosting Decision Trees (Li et al., 2020). We changed the strength of privacy protection using different values of `total_budget`: 100, 50, 10, and 5.

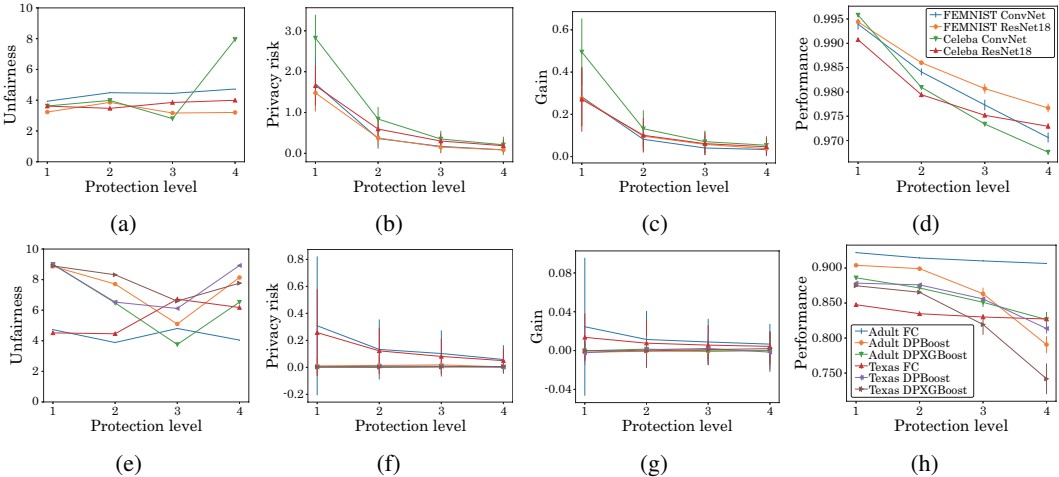

Figure 3: Main experimental results. (a) and (e) show unfairness when using the image and tabular datasets. (b)/(f) and (c)/(g) show means and standard deviations of users' privacy risks and gains. (d) and (h) show classification performance on data provided by users, i.e., training datasets. The classification performance (accuracy for multi-class classification and area under the curve (AUC) for binary classification) is calculated using 10 shadow models, and the mean and standard deviations are shown. The protection level $= [1, 2, 3, 4]$ corresponds to $\sigma = [0.1, 0.3, 0.5, 0.7]$ for NNs, `total_budget` $= [100, 50, 10, 5]$ for DPBoost, and `dp_epsilon_per_tree` $= [100, 10, 1, 0.1]$ for DPXGBoost.

- *DPXGBoost*: This is a differentially private XGBoost proposed for improving scalability (Grislain & Gonzalvez, 2021). We changed the strength of privacy protection using different values of `dp_epsilon_per_tree`: 100, 10, 1, and 0.1.

## 4.2 EXPERIMENTAL RESULTS

We evaluated the unfairness, privacy risks, gains, and classification performance changing the strength of privacy protection as shown in Fig. 3. When calculating the privacy risks using the NNs, we repeated the game shown in Fig. 7 in Appendix B 1,000 times to improve efficiency. When using the tree-based algorithms, we repeated the game shown in Fig. 1 1,000 times for each user. As detailed results, we show the distributions of $d_i$ in Fig. 8 in Appendix.

The results were different for NNs and tree-based algorithms. The results of NNs show that privacy risks, gains, and classification performance decreased as privacy protection became stronger. The unfairness of NNs was large without depending on the strength of privacy protection. These results show that we cannot improve the proposed IF by changing the strength of privacy protection. To look deeper into the results, we show users' privacy risks and gains in Fig. 4 when using ConvNet and $\sigma = 0.1$. The results show that privacy risks and gains of the majority of users had a positive correlation, but some users' gains were small compared with their risks. Such users were the main cause of unfairness. Figure 8 in Appendix shows such users were also the cause of unfairness in other settings.

The results of tree-based algorithms were surprising and show that privacy risks and gains were extremely small without depending on the strength of privacy protection. Unfairness was large without depending on the strength of privacy protection. Even though the unfairness of tree-based algorithms was large, users do not seem to consider that the unfairness is a problem because the magnitude of privacy risks and gains was similar and extremely small for all users. However, this result raises a new question about users' motivation for providing their data. For example, some users are unwilling to provide their data if their data do not improve the ML service.

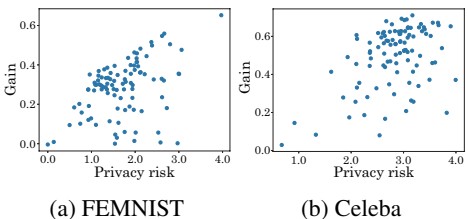

(a) FEMNIST    (b) Celeba

Figure 4: Privacy risks and gains of users when using ConvNet and $\sigma = 0.1$

Table 1: Changes in unfairness on FEMNIST

|  | FEMNIST/ConvNet | | | | FEMNIST/ResNet18 | | | |
| --- | --- | --- | --- | --- | --- | --- | --- | --- |
| $\sigma$ | 0.1 | 0.3 | 0.5 | 0.7 | 0.1 | 0.3 | 0.5 | 0.7 |
| Top1 | -1.02 | -1.13 | 0.74 | -0.14 | 0.02 | -0.88 | -0.17 | 2.68 |
| Top2 | -0.27 | -1.11 | 0.19 | 0.78 | 0.44 | -0.85 | -0.28 | 1.55 |
| Round1 | 0.01 | 0.89 | -0.73 | -0.36 | 1.11 | -0.82 | 0.72 | -0.15 |
| Round2 | 0.04 | 0.52 | -0.01 | -0.86 | 0.77 | -0.89 | 0.04 | 1.62 |
| Round3 | -0.00 | 0.65 | 0.47 | 0.13 | 0.43 | -1.15 | 0.30 | 1.64 |
| Temp5 | 0.09 | -0.58 | -1.09 | -0.90 | -0.08 | 1.17 | 2.24 | -0.22 |
| Temp15 | 0.30 | -0.58 | -1.58 | 1.75 | 0.19 | 0.35 | 1.97 | 2.62 |

Table 2: Changes in unfairness on Celeba, Adult, and Texas

|  | Celeba/ConvNet | | | | Celeba/ResNet18 | | | | Adult/FC | | | | Texas/FC | | | |
| --- | --- | --- | --- | --- | --- | --- | --- | --- | --- | --- | --- | --- | --- | --- | --- | --- |
| $\sigma$ | 0.1 | 0.3 | 0.5 | 0.7 | 0.1 | 0.3 | 0.5 | 0.7 | 0.1 | 0.3 | 0.5 | 0.7 | 0.1 | 0.3 | 0.5 | 0.7 |
| Round1 | 0.60 | 0.06 | -0.66 | -0.52 | 0.61 | 1.26 | -2.92 | -0.88 | -0.36 | -0.41 | -1.73 | -0.55 | 0.05 | 0.31 | 0.20 | -1.68 |
| Round2 | 0.70 | 0.42 | -0.74 | -0.77 | 0.61 | 0.80 | -3.92 | -1.27 | 1.82 | -0.17 | -0.75 | -0.04 | -1.04 | -0.72 | -1.10 | -1.65 |
| Round3 | 0.97 | 0.43 | -0.52 | -0.87 | 0.44 | 0.56 | -3.25 | 0.70 | 0.86 | -0.06 | -1.31 | 0.28 | -0.34 | -0.52 | -0.54 | -1.19 |
| Temp5 | 0.00 | -0.00 | 0.09 | -0.01 | 0.02 | -0.03 | -0.04 | 0.01 | -0.02 | 0.01 | -0.01 | -0.00 | 0.00 | -0.01 | -0.01 | -0.08 |
| Temp15 | 0.00 | -0.00 | 0.09 | -0.01 | 0.02 | -0.03 | -0.04 | 0.01 | -0.02 | 0.01 | -0.01 | -0.00 | 0.00 | -0.01 | -0.01 | -0.08 |

## 4.3 EFFECTIVENESS OF DEFENSES AGAINST MEMBERSHIP INFERENCE ATTACK

The experimental results of NNs indicate that the proposed IF can be improved by reducing the privacy risks of users whose gains are small. Hence, we further investigate the unfairness when applying defenses against membership inference attacks (MIAs). Since the proposed IF is important only for NNs based on the experimental results, we investigate changes in unfairness when applying defenses to NNs. In order not to decrease gains, we used three defenses that do not affect classification performance but can reduce privacy risks.

- Top$k$: This defense outputs predictions on $k$ classes with the largest confidence. This defense makes MIA difficult when the confidence of the correct class is small. We expect that this defense reduces the privacy risks of users whose accuracies are low. Note that this defense can be applied to only multi-class classification, i.e., FEMNIST. We used $k = 1, 2$.

- Round$d$: This defense rounds confidence to $d$ decimal places. This defense can make MIA difficult because attackers cannot use small changes in confidence. We used $d = 1, 2, 3$.

- Temp$t$: This defense replaces a temperature parameter of the softmax function with $t$. This defense can make MIA difficult because attackers cannot use small changes in confidence. We used $t = 5, 15$.

Tables 1–2 show changes in unfairness when applying the defenses. The negative and positive values mean that unfairness was decreased and increased by applying the defenses, respectively. We also show changes in privacy risks in Tables 6 in Appendix because the defenses affect privacy risks as well. The effects differ depending on defenses. Top$k$ improved the fairness when the privacy protection is relatively weak, i.e., $\sigma = 0.3$. Top1 and Top2 improved the fairness to a similar extent. Round$d$ improved the fairness when the privacy protection is strong, i.e., $\sigma = 0.5$ or $0.7$. Round1–3 improved the fairness to a similar extent except for FEMNIST and ConvNet. Round3 was not effective for FEMNIST and ConvNet. Temp$t$ improved the fairness of FEMNIST and ConvNet, but did not improve the fairness of the other settings. All defenses are effective in some settings, but no defense improved fairness in all settings. In contrast, unfairness was increased by the defenses in some settings. These results show that a method is required for improving the proposed IF of NNs.

## 5 DISCUSSION

**Necessity of the proposed individual fairness.** As aforementioned in Section 4.2, the proposed IF does not seem to be important for tree-based algorithms because both privacy risks and gains were extremely small. Here, we discuss the necessity of the proposed IF for NNs. The experimental results show that privacy risks decrease as the strength of privacy protection increases. With extremely

strong privacy protection, all users' privacy risks are expected to be negligibly small. In such a case, the proposed IF does not seem to be important. However, strong privacy protection for NNs deteriorates classification performance as shown in Fig. 3(d,h). For this reason, using an extremely large $\sigma$ is impractical. Using a moderate $\sigma$ that can limit privacy risks to an acceptable level and building an individually fair model is reasonable in practice.

**Possibility of other gains.** In this paper, we calculated gains by the accuracy improvement on users' data. However, the gains are not limited to the accuracy improvement, and we could assume other gains such as premium service and a monetary reward. Even when using such gains, we can evaluate IF based on our definition. Since all users need to obtain the gains corresponding to their risks to achieve small $\delta$ in our definition, the gains are required to be adjustable depending on users' risks.

# 6 RELATED WORK

**Fairness and privacy protection.** Related work regarding fairness and privacy protection is divided into three lines. The first and main line of work is studying the relation between group fairness and privacy protection. Many studies have investigated the effect of privacy protection with DP on group fairness (Bagdasaryan et al., 2019; Pujol et al., 2020; Farrand et al., 2020; Tran et al., 2021). All studies show that privacy protection deteriorates fairness. One study focused on the relationship between vulnerability against MIA and group fairness (Chang & Shokri, 2021). This study shows that there is a tradeoff between vulnerability and fairness because training data needs to be memorized to make a model fair.

The second line of work is studying the difference in privacy risks depending on groups (Zhang et al., 2020; Zhong et al., 2022). These studies employ MIA to evaluate privacy risks and show that minority groups face larger privacy risks than majority ones. The studies further show that privacy protection with DP decreases the difference in privacy risks between minority and majority groups.

The third line of work is proposing methods satisfying both privacy protection and group fairness. Xu et al. (2019) proposed a logistic regression guaranteeing both DP and group fairness, and Xu et al. (2020) extended DP-SGD for reducing unfairness by adjusting the clipping of each class.

**Lower bound of DP parameter $\epsilon$.** In this paper, when calculating the lower bound, we assume a realistic attacker who can access outputs of the trained model via an API. Not only such a realistic attacker but also stronger attackers were proposed to calculate lower bounds (Nasr et al., 2021). The stronger attackers can access intermediate models during training and manipulate the whole dataset. In this paper, we assume a realistic attacker because the trained models are assumed to be carefully protected by ML services.

# 7 CONCLUSION

In this paper, we propose a new definition of individual fairness (IF) from the perspective of privacy protection and experimentally evaluate privacy-preserving machine learning (ML) based on the proposed IF. For the proposed definition, we assume that users provide their data to an ML service and consider the principle that all users should obtain gains corresponding to their privacy risks. Furthermore, we discuss the relationship between the proposed IF and prior IF for classification and validate the proposed IF using synthetic data.

We conducted experiments on the image and tabular datasets using three neural networks (NNs) and two tree-based algorithms with differential privacy guarantee. The experimental results of NNs show that we cannot stably improve the proposed IF by changing the strength of privacy protection and applying defenses against membership inference attacks. These results show the need for a method that stably improves the proposed IF of NNs. The results of tree-based algorithms show that privacy risks and gains were extremely small without depending on the strength of privacy protection. For tree-based algorithms, the proposed IF seems not to be important, but these results raise a new question about the motivation of users for providing their data.

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

## A  PROOF OF PROPOSITION 1

*Proof.* From Definition 4, if a training algorithm satisfies $\delta$-IF, for a user $u_i$ having a privacy risk $r_i'$ and a gain $g_i'$, the following holds:

$$r_i' - \delta \le g_i' \le r_i' + \delta.$$

Unnormalizing $r_i'$ and $g_i'$ with $r_i' = \frac{r_i - \mu_r}{\sigma_r}$ and $g_i' = \frac{g_i - \mu_g}{\sigma_g}$, $r_i$ and $g_i$ satisfy the following:

$$\frac{\sigma_g}{\sigma_r}(r_i - \mu_r) - \sigma_g \delta + \mu_g \le g_i \le \frac{\sigma_g}{\sigma_r}(r_i - \mu_r) + \sigma_g \delta + \mu_g.$$

Considering a user $u_j$ having a privacy risk $r_j = r_i + \epsilon$, the gain $g_j$ satisfies the following:

$$\frac{\sigma_g}{\sigma_r}(r_i + \epsilon - \mu_r) - \sigma_g \delta + \mu_g \le g_j \le \frac{\sigma_g}{\sigma_r}(r_i + \epsilon - \mu_r) + \sigma_g \delta + \mu_g.$$

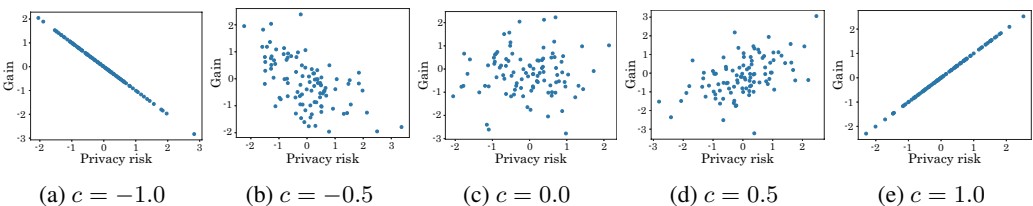

Figure 5: First type of synthetic data with different $c$

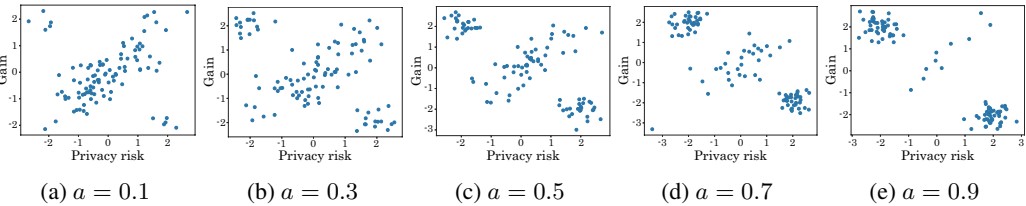

Figure 6: Second type of synthetic data with different $a$

The difference between gains of users $u_i$ and $u_j$ such that $d(r_i, r_j) = |r_i - r_j| \leq \epsilon$ takes the supremum when $r_j = r_i + \epsilon$ or $r_j = r_i - \epsilon$:

$$\max_{d(r_i, r_j) \leq \epsilon} |g_i - g_j| \leq \frac{\sigma_g}{\sigma_r}\epsilon + 2\sigma_g\delta = \delta'.$$

This shows that a training algorithm satisfies $\epsilon$-$\delta'$-IF. □

## B DETAILED EXPERIMENTAL SETUP

For the experiments of NNs, we used PyTorch v1.10.1 (Paszke et al., 2017) and Opacus v1.0.0 (Yousefpour et al., 2021) for implementation and conducted our experiments on NVIDIA Tesla V100 16GB with CUDA 11.3.1.

**Dataset.** We used two image datasets and two tabular datasets containing user information.

- *FEMNIST*: This dataset consists of 28×28 gray scale handwritten images written by 3,500 users (Caldas et al., 2018). We randomly selected 500 users from users having more than 100 images of digits 0–9. We used 59,556 images of the selected users as the training dataset $D_0$ and trained models for 10-class classification.
- *Celeba*: This dataset consists of images of 9,343 users for facial expression recognition (Caldas et al., 2018). We resized images to $64 \times 64$ and randomly selected 2,000 users from users having more than 10 images. We used 46,116 images of the selected users as $D_0$ and trained models classifying "smile" and "not smile".
- *Adult*: This dataset consists of 14 types of users' attributes extracted from the 1994 US Census database (Kohavi & Becker, 1996). We used one-hot encoding for categorical attributes, and the size of encoded feature vectors is 108. We randomly selected 39,073 users as $D_0$ and trained models classifying whether a user's income is larger than 50k.
- *Texas*: This dataset consists of patients' attributes (Texas Department of State Health Services, 2013). We used 12 types of attributes: `DISCHARGE`, `TYPE_OF_ADMISSION`, `PAT_STATE`, `PAT_STATUS`, `SEX_CODE`, `RACE`, `ETHNICITY`, `ADMIT_WEEKDAY`, `PAT_AGE`, `RISK_MORTALITY`, `ILLNESS_SEVERITY`, `LENGTH_OF_STAY`. We used one-hot encoding for categorical attributes, and the size of encoded feature vectors is 72. We randomly selected 50,000 users as $D_0$ and trained models classifying whether a patient's `TOTAL_CHARGES` is larger than 50k.

We normalized each channel of FEMNIST and Celeba so that its mean and variance are 0 and 1, respectively. We normalized each element of Adult and Texas so that its minimum and maximum values are 0 and 1, respectively.

Table 3: Hyperparameters

| Algorithm | Hyperparameter | Candidates | Selected |
|---|---|---|---|
| ConvNet | Patch size | $3 \times 3, 5 \times 5$ | $5 \times 5$ |
| | Output channel size | 32, 64, 128 | 64 |
| | Number of units in the FC layer | 256, 512, 1024 | 512 |
| | Batchsize | 64, 128, 512 | 512 |
| FC | Number of layers | 3, 5, 10 | 5 |
| | Number of units | 100, 500, 1,000 | 500 |
| DPBoost | num_leaves | 20, 30, 50, 100 | 30 |
| | max_depth | 3, 6, 10 | 10 |
| | my_n_trees | 20, 30, 50 | 50 |
| | inner_boost_round | 20, 30, 50 | 50 |
| DPXGBoost | n_trees | 10, 20, 30 | 20 |
| | subsample | 0.1, 0.2, 0.3 | 0.2 |
| | max_depth | 3, 6, 10 | 6 |

Table 4: Architecture of ConvNet

| Layer | Type (Activation) | Patch | Output channels |
|---|---|---|---|
| 1 | Convolution (ReLU) | $5 \times 5$ | 64 |
| 2 | Max pooling | $2 \times 2$ | 64 |
| 3 | Convolution (ReLU) | $5 \times 5$ | 64 |
| 4 | Max pooling | $2 \times 2$ | 64 |
| 5 | Fully connected (ReLU) | | 512 |
| 6 | Fully connected (Softmax) | | 10 or 2 |

For each dataset, we selected 100 users from the test datasets to evaluate fairness. Even though 100 users are a part of the users, we can estimate the overall trend of the proposed IF by selecting users considering their variety. To select a variety of users from ordinary to outlier users, we trained 10 shadow models using the training dataset and calculated the accuracy of each user's data. We sorted users in descending order, selected 20 users each from $\frac{4}{8} \times 100$, $\frac{5}{8} \times 100$, $\frac{6}{8} \times 100$, and $\frac{7}{8} \times 100$ percentiles, and selected 20 users with the lowest accuracies. Since each user has multiple data in image datasets, we selected data with the lowest accuracy from each user's data for the evaluation. We calculated the accuracy using ConvNet (described below) and $\sigma = 0.1$. Note that each user has one sample of data in tabular datasets.

**Machine learning algorithm and hyperparameters.** We used three NNs and two tree-based algorithms. We selected the best hyperparameters in terms of classification performance on the test dataset. For classification performance, we calculate accuracy for multi-class classification, i.e., FEMNIST, and AUC for binary classification, i.e., Celeba, Adult, and Texas. The candidates of hyperparameters and selected ones are shown in Table 3. We used different random seeds for each experiment and set a millisecond obtained with `time.time()` as a random seed.

- *ConvNet*: This is a six-layer CNN designed by referring to the prior work (Nasr et al., 2021). ConvNet consists of two convolutional layers, two max-pooling layers, and two fully connected layers. The details of the architecture are shown in Table 4. We optimize ConvNet with DP-SGD using commonly used hyperparameters; the number of epochs is 50, the batchsize is 512, the optimizer is Adam (Kingma & Ba, 2014), the learning rate $\eta$ is 0.001, the clipping threshold of the gradient $C = 1$, and the privacy parameter $\delta = 10^{-5}$. We changed the strength of privacy protection using different variances of the noises: $\sigma = 0.1, 0.3, 0.5$, and $0.7$. The corresponding upper bounds of $\epsilon$ by theoretical analysis are 42,967.4, 217.0, 36.1, and 12.5.

- *ResNet18*: This is a 18-layer CNN with the shortcut connections (He et al., 2016). To apply DP-SGD to ResNet18, we replaced the BatchNorm layers with GroupNorm layers referring to the tutorial of Opacus (Meta Platforms, Inc., 2022). BatchNorm layers cause a privacy violation because they use means and variances regarding samples in a minibatch.

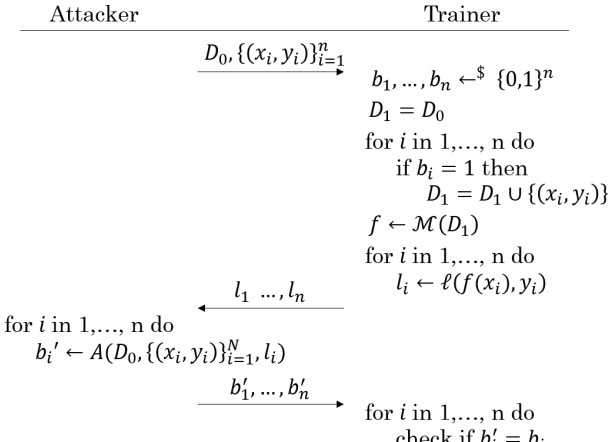

Figure 7: Multiple-sample game for efficient calculation of lower bound

The means and variances make dependencies between samples in a minibatch and violate DP. For this reason, we fix ResNet18 with `ModuleValidator`. For optimization, we used the DP-SGD and the same hyperpatramers as ConvNet.

- *FC*: This is a five-layer fully connected neural network. The number of units in the intermediate layers is 500, and their activation is ReLU. For optimization, we used the DP-SGD and the same hyperparameters as ConvNet.

- *DPBoost*: This is a differentially private Gradient Boosting Decision Trees (Li et al., 2020). DPBoost obtains a tighter sensitivity bound with Gradient-based Data Filtering and Geometric Leaf Clipping. We changed the strength of privacy protection by specifying the upper bounds of $\epsilon$ using different values of `total_budget`: 100, 50, 10, and 5.

- *DPXGBoost*: This is a differentially private XGBoost proposed for improving scalability (Grislain & Gonzalvez, 2021). We changed the strength of privacy protection using different values of `dp_epsilon_per_tree`: 100, 10, 1, and 0.1. The corresponding upper bounds of $\epsilon$ by theoretical analysis are 1,967.8, 167.8, 5.9, and 0.4.

**Efficient calculation of lower bound.** One drawback of the lower bound calculation is computational cost. Specifically, when the game is repeated 1,000 times, 1,000 models need to be trained with privacy-preserving ML in total. In our experiments, we calculate the lower bounds for 100 users and need to train $100 \times 1,000$ models. When using NNs, the computational cost is extremely expensive. To tackle this problem, we use a multiple-sample game shown in Fig. 7 designed by referring to an efficient method for label DP (Malek Esmaeili et al., 2021). In the multiple-sample game, the attacker sends a dataset $D_0$ and $n$ target samples $\{(\boldsymbol{x}_i, y_i)\}_{i=1}^n$ to the trainer. The trainer randomly decides whether to use each target sample for training. In Fig. 7, $b_i$ represents whether the sample $(\boldsymbol{x}_i, y_i)$ is used for training. The trainer builds a model using the dataset $D_0$ and the selected target samples with a privacy-preserving ML algorithm $\mathcal{M}$ and returns the losses on the target samples to the attacker. The attacker predicts whether each sample is used for training with the dataset $D_0$, the target samples $\{(\boldsymbol{x}_i, y_i)\}_{i=1}^n$, and the loss $l_i$ on the sample $(\boldsymbol{x}_i, y_i)$. Finally, the trainer checks if the predictions are correct. We repeat the multiple-sample game multiple times and calculate the lower bound $\epsilon_{LB}$ following Eq. 1 for each target sample. When we use 100 samples in our game, we can reduce the number of training by 1/100.

We validated the reliability of the multiple-sample game using users' data with the lowest accuracies. We calculated the lower bounds with the game shown in Fig. 1 and the multiple-sample game shown in Fig. 7 and checked if the lower bounds were close to each other. We repeated both games 1,000 times each using DP-SGD parameter $\sigma = 0.1$ and used 100 target samples in the multiple-sample game. Table 5 shows that we obtained good approximations of the lower bounds with the multiple-sample game.

Table 5: Lower bounds of $\epsilon$ calculated with one-sample and multiple-sample games. The one-sample game is a game shown in Fig. 1.

| Dataset | Network | One-sample | Multiple-sample |
|---------|---------|------------|-----------------|
| FEMNIST | ConvNet | 2.00 | 1.98 |
| FEMNIST | ResNet18 | 2.99 | 2.58 |
| Celaba | ConvNet | 3.21 | 3.35 |
| Celaba | ResNet18 | 2.53 | 2.35 |
| Adult | FC | 1.73 | 1.52 |
| Texas | FC | 1.00 | 1.05 |

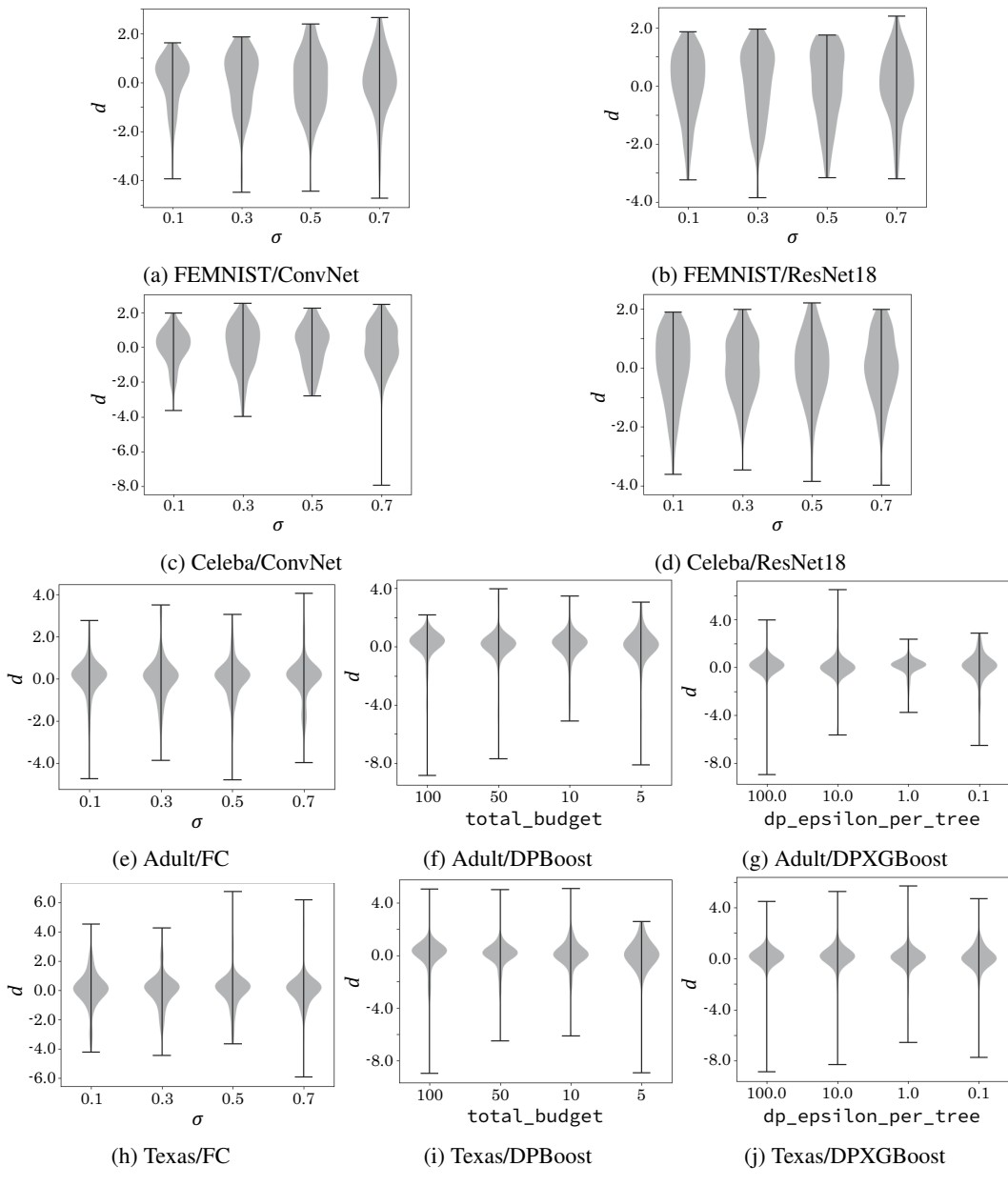

Figure 8: Distribution of $d_i$

Table 6: Changes in privacy risks when applying defenses. We calculated a change for each user and show the means and standard deviations of the changes. The negative and positive values mean that privacy risks were decreased and increased by applying the defenses, respectively.

| | FEMNIST/ConvNet | | | | FEMNIST/ResNet18 | | | |
|---|---|---|---|---|---|---|---|---|
| $\sigma$ | 0.1 | 0.3 | 0.5 | 0.7 | 0.1 | 0.3 | 0.5 | 0.7 |
| Top1 | -0.58 ± 0.55 | -0.17 ± 0.21 | -0.09 ± 0.11 | -0.06 ± 0.11 | -0.40 ± 0.42 | -0.15 ± 0.19 | -0.09 ± 0.11 | -0.06 ± 0.09 |
| Top2 | -0.30 ± 0.46 | -0.08 ± 0.20 | -0.06 ± 0.11 | -0.03 ± 0.08 | -0.16 ± 0.22 | -0.07 ± 0.14 | -0.05 ± 0.11 | -0.04 ± 0.08 |
| Round1 | -0.64 ± 0.52 | -0.19 ± 0.19 | -0.13 ± 0.14 | -0.07 ± 0.10 | -0.39 ± 0.28 | -0.16 ± 0.17 | -0.09 ± 0.10 | -0.06 ± 0.09 |
| Round2 | -0.51 ± 0.48 | -0.16 ± 0.18 | -0.09 ± 0.11 | -0.06 ± 0.10 | -0.25 ± 0.24 | -0.11 ± 0.16 | -0.07 ± 0.10 | -0.05 ± 0.08 |
| Round3 | -0.42 ± 0.47 | -0.12 ± 0.17 | -0.08 ± 0.10 | -0.05 ± 0.09 | -0.15 ± 0.21 | -0.07 ± 0.13 | -0.05 ± 0.09 | -0.04 ± 0.07 |
| Temp5 | -0.02 ± 0.11 | 0.00 ± 0.06 | -0.00 ± 0.08 | 0.00 ± 0.06 | -0.01 ± 0.19 | 0.01 ± 0.10 | 0.00 ± 0.11 | -0.02 ± 0.06 |
| Temp15 | -0.13 ± 0.26 | -0.02 ± 0.10 | -0.03 ± 0.12 | -0.01 ± 0.10 | -0.12 ± 0.26 | -0.04 ± 0.15 | -0.00 ± 0.15 | -0.03 ± 0.10 |
| | Celeba/ConvNet | | | | Celeba/ResNet18 | | | |
| $\sigma$ | 0.1 | 0.3 | 0.5 | 0.7 | 0.1 | 0.3 | 0.5 | 0.7 |
| Round1 | -0.84 ± 0.48 | -0.53 ± 0.47 | -0.33 ± 0.34 | -0.27 ± 0.23 | -0.17 ± 0.15 | -0.15 ± 0.14 | -0.15 ± 0.20 | -0.12 ± 0.12 |
| Round2 | -0.67 ± 0.45 | -0.38 ± 0.41 | -0.23 ± 0.31 | -0.20 ± 0.23 | -0.13 ± 0.15 | -0.12 ± 0.13 | -0.11 ± 0.18 | -0.09 ± 0.11 |
| Round3 | -0.51 ± 0.40 | -0.20 ± 0.22 | -0.16 ± 0.25 | -0.12 ± 0.17 | -0.11 ± 0.14 | -0.07 ± 0.11 | -0.08 ± 0.18 | -0.06 ± 0.09 |
| Temp5 | -0.00 ± 0.01 | -0.00 ± 0.01 | 0.00 ± 0.01 | 0.00 ± 0.01 | 0.00 ± 0.00 | 0.00 ± 0.01 | 0.00 ± 0.01 | -0.00 ± 0.00 |
| Temp15 | -0.00 ± 0.01 | -0.00 ± 0.01 | 0.00 ± 0.01 | 0.00 ± 0.01 | 0.00 ± 0.00 | 0.00 ± 0.01 | 0.00 ± 0.01 | -0.00 ± 0.00 |
| | Adult/FC | | | | Texas/FC | | | |
| $\sigma$ | 0.1 | 0.3 | 0.5 | 0.7 | 0.1 | 0.3 | 0.5 | 0.7 |
| Round1 | -0.22 ± 0.41 | -0.08 ± 0.16 | -0.08 ± 0.14 | -0.05 ± 0.09 | -0.20 ± 0.30 | -0.10 ± 0.15 | -0.07 ± 0.12 | -0.04 ± 0.08 |
| Round2 | -0.18 ± 0.38 | -0.07 ± 0.16 | -0.06 ± 0.12 | -0.04 ± 0.08 | -0.16 ± 0.31 | -0.08 ± 0.14 | -0.06 ± 0.11 | -0.04 ± 0.07 |
| Round3 | -0.08 ± 0.17 | -0.04 ± 0.10 | -0.04 ± 0.12 | -0.03 ± 0.07 | -0.08 ± 0.17 | -0.04 ± 0.10 | -0.03 ± 0.06 | -0.03 ± 0.06 |
| Temp5 | -0.00 ± 0.00 | -0.00 ± 0.01 | 0.00 ± 0.00 | 0.00 ± 0.00 | 0.00 ± 0.01 | -0.00 ± 0.01 | -0.00 ± 0.00 | -0.00 ± 0.00 |
| Temp15 | -0.00 ± 0.00 | -0.00 ± 0.01 | 0.00 ± 0.00 | 0.00 ± 0.00 | 0.00 ± 0.01 | -0.00 ± 0.01 | -0.00 ± 0.00 | -0.00 ± 0.00 |

