# OpenReview forum: "Individual Fairness of Data Provider Regarding Privacy Risk and Gain"
_ICLR.cc/2023/Conference — Submitted to ICLR 2023_

### Official Review · Reviewer_bzAv · 2022-10-20

**Confidence:** 4
**Correctness:** 3
**Technical Novelty And Significance:** 2
**Empirical Novelty And Significance:** 3
**Recommendation:** 5

**Clarity, Quality, Novelty And Reproducibility:**

__Clarity__: The paper is well-written and easy to follow. Mathematical notation is well described, and connections between sections make sense. Overall, the paper is a good read.

__Quality and Novelty__: The quality of work is unquestionably well, and analyses that bring these privacy-preserving approaches closer to practical scenarios are much needed. Although the approach of measuring utility by simply adding datapoints is not novel (and faster approximations such as influence function-based methods do exist, especially in model poisoning literature), focus of the privacy lens to potential participants is new.

__Reproducibility__: Detailed experimental setups are described in the paper (and in detail in the Appendix). Although minor details (like exact seeds) are not available to perfectly reproduce results, sufficient information is provided to achieve results with expected trends.

### Minor comments

- Page 1: "...users decide whether to provide their data to ML service." Can the authors please provide concrete real-world examples? It is hard to think of any instances where the users have a clear choice like this and is directly impacted by that decision.
- Figure 1 is not a Figure- please rename the type.
- Section 3.2: "...instead of the loss decrease because..." model users may care about metrics like precision/recall/F-1 which are computed for specific thresholds, the performance of which can be impacted by different loss values indeed. The reasoning here to use accuracy is not very convincing.
- Section 4.2: "...changing the strength of privacy protection." Please add a more explicit discussion on 'how' this varying strength of privacy protection is achieved.
- Section 4.3 argues that Topk "makes MIa difficult when the confidence of the correct class is small", which is unlikely for a well-performing classifier to begin with.

**Strength And Weaknesses:**

## Strengths

- This work introduces the user's perspective (which is the data owner, and should thus be at the center of decisions in an ideal scenario) to privacy-preserving learning- convincing users to participate in training, and in some way demonstrating benefits to privacy and/or utility. Although the results are negative, they show how MIA defenses and increasingly private learning do not necessarily lead to improved performance for 'all' users, which is what model trainers should strive for.

- Evaluation is extensive, and the notion of $(\epsilon, \delta)$-IF (and its connection to IF) is well explained and intuitive to follow and understand.

## Weaknesses

- The authors acknowledge that there "are differences in privacy risks between users", and argue that a training algorithm is IF if all users gain gains corresponding to privacy risks. However, this seems to defeat the very purpose of privacy and algorithmic fairness in general. All users' privacy guarantees should be similar, and not unequally biased towards a specific sub-group of users; the same goes for model performance as well. From what I understand, IF says 'if you had poor privacy guarantees, you should be content with this gain, even though it may be less of a gain than some other user who already had better privacy guarantees. I am not convinced that this is the right approach to dealing with privacy, and is certainly not fair.

- The proposed approach seems to be excessively computationally expensive. Even for a relatively small shadow model, re-training for each potential user is not feasible at all (and opens doors to other kinds of adversaries that may craft adversarial data and use the "expected gains" to leak sensitive information from the victim's models/training data). If a user is wary of the model trainer in the first place (which is why they want to look at potential gains), how can the same user be trusted with any 'metrics' that the model trainer may report as expected gains? One way could be to provide black-box API access to the trained model as proof, but that itself is a security/privacy risk.

- Although this is a start, true overlook of "individual" fairness should really look at "individuals", not their individual records. A user, for instance, may have 5-10 images that they might upload to a cloud provider. Although the privacy/performance gains would be low when measured for a single data point (which is expected, given the relative scale of training data), they would be much higher if a collection of data is used instead. With the help of subject-level membership attacks [1, 2], similar gains can be measured for this case as well, which is much better aligned with real-world use cases.

- I would like to see the use of some other defenses already proposed in the literature, in addition to the ones described in Section 4.3. For instance, causal learning is known to reduce inference risk significantly [3].

### References

[1] Hartmann, V., Meynent, L., Peyrard, M., Dimitriadis, D., Tople, S., & West, R. (2022). Distribution inference risks: Identifying and mitigating sources of leakage. arXiv preprint arXiv:2209.08541.

[2] Suri, A., Kanani, P., Marathe, V. J., & Peterson, D. W. (2022). Subject Membership Inference Attacks in Federated Learning. arXiv preprint arXiv:2206.03317.

[3] Tople, S., Sharma, A., & Nori, A. (2020, November). Alleviating privacy attacks via causal learning. In International Conference on Machine Learning (pp. 9537-9547). PMLR.

**Summary Of The Paper:**

This work proposes a new notion of privacy, specifically targeted at individuals, rather than the entire training dataset as a whole. The core idea behind the concept of IF is that individuals should get gains (in privacy and performance) out of participation. Most results highlight disturbing trends in individual fairness- specifically for neural networks. Although most of the results (vis-a-vis tree algorithms and MIA defenses) are negative in nature, they raise the essential question of why users should participate in such model training routines, given their performance and privacy gains.

**Summary Of The Review:**

The main issue with the paper in its current form is how it convinces the reader about the severity of this issue. Particularly, the approach of retraining models (even if shadow) for every potential participant is far from feasible, and it is unclear if relative gains should be the focus of the study, given the aspect of fairness (eliminating propagation of biases, in the form of privacy or performance gains). I think if the authors can address this main concern (discussed above) and add comparisons with some defenses (like causal learning, or something apart from Differential Privacy that is shown to work well in the literature), the paper would be in much better shape.

---

### Official Review · Reviewer_fzWY · 2022-10-25

**Confidence:** 4
**Correctness:** 3
**Technical Novelty And Significance:** 2
**Empirical Novelty And Significance:** 2
**Recommendation:** 3

**Clarity, Quality, Novelty And Reproducibility:**

**Clarity** The paper is well-written and clearly organized. However, I think some sections can be moved to the appendix. For example, though the synthetic dataset section is interesting, the section does not contribute a lot to the main story.

**Quality** The submission seems incomplete, and there is a large room for improvement. For example, no method is proposed for the new IF notion, and the empirical findings do not provide any valuable new insights.

**Novelty** The proposed new IF is novel but needs more justification.

**Reproducibility** No issues with the reproducibility if the author could release the codes.

**Strength And Weaknesses:**

Strengths:
1. The new IF notion is novel in terms of its formulation.
2. The empirical study is thorough regarding the diversity of model architecture, datasets, etc.

Weaknesses:
1. The contributions of the paper are limited by the motivation of the proposed IF. First of all, the authors do not justify the proposed IF well, and the requirement that similar gain corresponds to similar privacy is arguable. I suggest the author better justify the proposed IF notions by several practical use cases.
2. Even though the definition makes sense, the paper does not provide any methods to mitigate the unfairness. It is not surprising that existing privacy protection methods do not provide any useful mitigation to the fairness problem. If the authors could provide some mitigation methods to the proposed fairness notion, even though it is only empirical,
3. Lastly, the empirical findings do not provide additional insights to the community: for the tree model, the proposed IF notion is unnecessary; For the NN models, it has been well known that the majority of existing DP methods cause significant performance loss.


**Summary Of The Paper:**

The paper proposed a new definition of individual fairness (IF), which requires all users’ tradeoffs between gain and privacy risks are similar. Empirical analysis shows different findings of how the fairness of neural networks and tree-based algorithms change under different privacy protections.

**Summary Of The Review:**

The contributions of the paper are limited by the motivation of the proposed IF, the lack of mitigation methods for unfairness, and the limited values provided by the empirical findings.

---

### Official Review · Reviewer_77T3 · 2022-10-25

**Confidence:** 4
**Correctness:** 2
**Technical Novelty And Significance:** 2
**Empirical Novelty And Significance:** 2
**Recommendation:** 3

**Clarity, Quality, Novelty And Reproducibility:**

The writing could be improved a lot. I think perhaps the most important part, where the technical contributions lies,
are the two dense paragraphs at the end of page 4. These should be expanded, it is difficult to find the definitions
of $g_i$'s and $r_i$'s from there.

- p.3: "upper bounds of FPR and FNR calculated with the Clopper-Pearson method..."
This should be elaborated, I don't think it is enough to simply cite here a paper.


**Strength And Weaknesses:**

Pros:

- The problem the paper is studying is very interesting, this is a timely problem.

- The main idea is interesting: "We consider that a training algorithm is individually fair if all users obtain gains corresponding to their privacy risks."


Cons:

- The content is a bit thin. There are only empirically computable quantities defined, which are tested in experiments.

- The paper contains some errors. For example:

"A training algorithm is individually fair if the difference in privacy risks between any pair of users is small. This definition can be satisfied by reducing privacy risks with privacy-preserving ML because the differences are small if all users’ privacy risks are small. "

I would claim this is not true. As shown e.g. in [1] and [2], there can be very large differences in individual DP epsilons between
users.


**Summary Of The Paper:**

The paper considers a timely topic: connection between fairness and privacy-preservation of an ML model. The angle is interesting: fairness is considered from the point of view that the model performance gains for user's data should correlate with the privacy leakage. It is well known that e.g. training with differential privacy leads to unequal individual privacy-preservation, see e.g. [1] and [2]. It is also known that the model performance correlates with privacy-preservation in a sense that the most accurate classes have best privacy-protection [2].

This paper consider empirical epsilons instead of a priori DP epsilons, computed as in [3] using a membership attack guessing game. These empirical epsilons are compared to empirical 'gain values' computed with models that contain and do not contain the particular sample. These (normalized) empirical values are used to determine the fairness and this fairness definition is tested on several experiments.


[1] Feldman, V., & Zrnic, T. (2021). Individual privacy accounting via a renyi filter. Advances in Neural Information Processing Systems, 34, 28080-28091.

[2] Yu, D., Kamath, G., Kulkarni, J., Yin, J., Liu, T. Y., & Zhang, H. (2022). Per-Instance Privacy Accounting for Differentially Private Stochastic Gradient Descent. arXiv preprint arXiv:2206.02617.

[3] Nasr, M., Songi, S., Thakurta, A., Papernot, N., & Carlin, N. (2021, May). Adversary instantiation: Lower bounds for differentially private machine learning. In 2021 IEEE Symposium on security and privacy (SP) (pp. 866-882). IEEE.

**Summary Of The Review:**

All in all I don't think the paper is mature for publication. The novelty and content is not enough and also the paper requires some rewriting in my opinion.

Instead of only comping up with a new definition of fairness, it would be nice if some remedies were suggested. As for example shown already in [2], it happens to be so that there is a correlation between privacy preservation and model performance: the classes with the lowest average epsilons have also the best performance. And this seems to be inherent to DP-SGD trained models (and probably to DP-trained ML models in general). So instead of inventing new tools to study the individual fairness, it might be more interesting to try find remedies. And this, as far as I see, this paper is not able to do.

---

### Official Review · Reviewer_MgQj · 2022-10-26

**Confidence:** 3
**Correctness:** 3
**Technical Novelty And Significance:** 2
**Empirical Novelty And Significance:** 3
**Recommendation:** 5

**Clarity, Quality, Novelty And Reproducibility:**

The paper is clearly written and reads well, however, the novelty and the new insights provided by it are limited. The results seem reproducible.

**Strength And Weaknesses:**

Strengths
1. The paper is looking at individual fairness which is an understudied problem. They consider the interplay of individual accuracy increase and privacy gain which is new.

Weaknesses
1. I am not entirely sure about the use of individual training accuracy gain as part of the metric, I think we should technically look at user-level accuracy, where we have at least a single sample held out test for a given training sample thereby measuring generalization accuracy gains. I think measuring the gain on training samples is a bit meaningless as technically the default accuracy on those samples could be 100% for that user, if we do like a KNN classifier.

2. The measurement of privacy expenditure is inconsistent, there should just be 4 privacy budget epsilons and all experiments done on those, as opposed to different epsilons and even different ways of reporting. Like for one method epsilon is reported,for another the standard deviation of the noise added is reported (which could be converted to epsilon and I think should be for presenting results). All this said, I find stacking all these inconsistent guarantees together like that in Figure 3 in appropriate.

3. I am not sure I fully understand the two lines of reasoning for why the naive definitions of IF are bad, in section 3.2:
•	A training algorithm is individually fair if users having similar data face similar privacy risks. This definition is inadequate because a difference in privacy risks can be large if users’ data are dissimilar --> I am not sure how this relates to the former sentence. Doesn't really make sense.
•	A training algorithm is individually fair if the difference in privacy risks between any pair of users is small. This definition can be satisfied by reducing privacy risks with privacy-preserving ML because the differences are small if all users’ privacy risks are small. However, strong privacy protection with DP is known to degrade classification performance --> There are two issues with this: a) the only way to get similar privacy risks is not by applying privacy preserving methods, there could be other ways out there, so to say that this is achieved by strong DP guarantees is a bit inaccurate, and b) there are recent papers that show DP can be achieved with little loss to accuracy [3-5]


4. I am not really sure what the conclusion  “necessity of the proposed IF for NNs'' really means. If it means we need improvements, there are some group level improvements, why not try them and then test? Improvments like [1-2]. I think the fact that the only conclusion from the paper is that individual level fairness is not good in privacy preserving methods is a bit repetitive gien prior work.

[1]  Tran, Cuong, Ferdinando Fioretto, and Pascal Van Hentenryck. "Differentially private and fair deep learning: A lagrangian dual approach." Proceedings of the AAAI Conference on Artificial Intelligence. Vol. 35. No. 11. 2021.

[2] Fioretto, Ferdinando, et al. "Differential Privacy and Fairness in Decisions and Learning Tasks: A Survey." arXiv preprint arXiv:2202.08187 (2022).

[3] Li X, Tramer F, Liang P, Hashimoto T. Large language models can be strong differentially private learners. arXiv preprint arXiv:2110.05679. 2021 Oct 12. (ICLR 2022)

[4] Yu D, Naik S, Backurs A, Gopi S, Inan HA, Kamath G, Kulkarni J, Lee YT, Manoel A, Wutschitz L, Yekhanin S. Differentially private fine-tuning of language models. arXiv preprint arXiv:2110.06500. 2021 Oct 13. (ICLR 2022)

[5] Tramer, Florian, and Dan Boneh. "Differentially Private Learning Needs Better Features (or Much More Data)." International Conference on Learning Representations. 2020.

**Summary Of The Paper:**

This paper works on the problem of individual fairness in terms of privacy risks and improved accuracy. More concretely, differential privacy has been shown to have a disparate impact on model accuracy for minorities, meaning groups with fewer members/outliers tend to lose more accuracy (by use of DP) than other larger groups. What this paper does is study this phenomenon, on an individual level, i.e. how much each individual loses, as opposed to each group. To enable this, the authors propose a new individual fairness metric that takes into account both the privacy risk and utility gain. They basically measure how much accuracy gain a single sample gets by participating in training, versus how much privacy they lose in terms of membership inference power.
They show that the shared trend is the more privacy risk, the better accuracy, and that similar to what was priorly shown on a group level, applying DP and other privacy mitigation attempts has a disparate impact on gains on an individual level.

**Summary Of The Review:**

I have provided an extended review in the strengths and weaknesses box. To summarize, I think the problem the authors are studying is important and interesting, and the proposed individual fairness measure seems new. However, what I am skeptical about is the novelty of the findings, as it seems to mostly reflect the disparate impact of privacy-preserving methods which has been shown before on multiple different setups, and there are mitigations provided for it too [1-2]. Given all this, I find the contributions limited. However, there is a chance that there is something I am missing, so if the authors clarify their contributions and I realize I have misunderstood something I am willing to increase my score.

---

### Decision · Program_Chairs · 2023-01-20

**Decision:**

Reject

**Justification For Why Not Higher Score:**

The article needs improvement before it can be presented at a major venue.


**Justification For Why Not Lower Score:**

N/A

**Metareview: Summary, Strengths And Weaknesses:**

This paper proposes an interesting idea: viewing individual fairness with respect to privacy. In particular, they want to define an algoritm as individually fair "if all users obtain gains corresponding to their privacy risks". This notion is not very well-defined in the introduction. For example, the authors say 'We estimate the accuracy improvement using shadow models" but not even a reference provided as to what that might mean. Overall, the paper could benefit from some crisper definitions.

The reviewers generally agree that for this, and other reasons, the paper does not clear the bar, but can certainly be improved for a subsequent submission.